# Generating Extractive Answers: Gated Recurrent Memory Reader for Conversational Question Answering

**Xuanyu Zhang**
Du Xiaoman Financial
Beijing, China
zhangxuanyu@duxiaoman.com

**Qing Yang**
Du Xiaoman Financial
Beijing, China
yangqing@duxiaoman.com

## Abstract

Conversational question answering (CQA) is a more complicated task than traditional single-turn machine reading comprehension (MRC). Different from large language models (LLMs) like ChatGPT, the models of CQA need to extract answers from given contents to answer follow-up questions according to conversation history. In this paper, we propose a novel architecture, i.e., Gated Recurrent Memory Reader (GRMR), which integrates traditional extractive MRC models into a generalized sequence-to-sequence framework. After the passage is encoded, the decoder will generate the extractive answers turn by turn. Different from previous models that concatenate the previous questions and answers as context superficially and redundantly, our model can use less storage space and consider historical memory deeply and selectively. Experiments on the Conversational Question Answering (CoQA) dataset show that our model achieves comparable results to most models with the least space occupancy.

## 1 Introduction

Recently, large language models (LLMs) like ChatGPT (OpenAI, 2022) and GPT-4 (OpenAI, 2023) have revolutionized the question-answering and conversation domains, pushing them to new heights. Different from ChatGPT-style conversations, the task of conversational question answering (CQA) requires models to answer follow-up questions using extracted answers based on given passages and conversation history. It can be regarded as an expansion of traditional single-turn machine reading comprehension (MRC) to multi-turn conversations. However, the follow-up questions usually have more complicated phenomena, such as co-reference, ellipsis and so on. It is necessary to consider historical memory in a conversation. To enable machines to answer such questions, many CQA datasets, such as CoQA

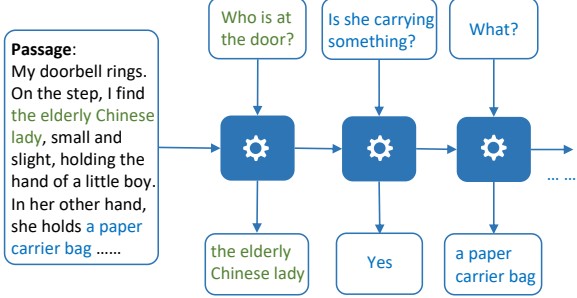

Figure 1: An example from CoQA.

(Reddy et al., 2018), QuAC (Choi et al., 2018) and QBLink (Elgohary et al., 2018), are proposed. Generally, a dialogue contains a long passage and some short questions about this passage. The current question may rely on previous questions or answers. Here is an example from CoQA dataset in Figure 1. Only if we know the conversation history can we understand that the third question *"What?"* represents *"What is she carrying?"*.

However, most previous approaches view this task as a traditional single-turn MRC task by concatenating previous questions or answers as conversation history superficially, such as BiDAF++ (Yatskar, 2018), DrQA+PGNet (Reddy et al., 2018), SDNet (Zhu et al., 2018) and so on. They can not grasp and understand the representation of history profoundly. And multiple historical questions and answers in one sentence may confuse the model.

Besides, these methods occupy a lot of storage space or precious and limited graphics memory because of duplicated questions and passages. Although, FlowQA (Huang et al., 2018) proposes a flow mechanism without concatenating previous questions, the hidden states of the passage are still duplicated many times to obtain the question-aware passage for each question. Meanwhile, it can not utilize dialogue history selectively.

To address these issues, we propose our new

architecture, i.e., **G**ated **R**ecurrent **M**emory **R**eader (GRMR). It integrates traditional MRC models into a generalized sequence-to-sequence (seq2seq) (Sutskever et al., 2014) framework to generate extractive answers. For one thing, the deeper representation of historical conversation can be utilized selectively by our recurrent structure and gated mechanism. For another thing, our model uses original questions and passages without redundancy, which saves lots of storage and memory. Experiments on the Conversational Question Answering (CoQA) dataset show that our model achieves comparable results to most models with the least space.

## 2   Related Work

In the field of generative conversations, large language models have become the prevailing approach. These models (OpenAI, 2022, 2023; Zhang and Yang, 2023b) are typically pretrained using vast amounts of unsupervised text data and subsequently fine-tuned using supervised instruction data. This supervised instruction data is often obtained through human annotations or distilled from existing large-scale models (Zhang and Yang, 2023a).

In the field of extractive conversations, the answers are typically extracted directly from the original passages by the model. This task is often seen as an extension of single-turn machine reading comprehension to multi-turn conversations. Prior to the era of pretraining, attention mechanisms were commonly employed in various subdomains of question answering, such as BiDAF (Seo et al., 2017) and Rception (Zhang and Wang, 2020) in classic single-turn machine reading comprehension, BiDAF++ (Yatskar, 2018) in conversational machine reading comprehension, and other methods based on multi-modal or structured knowledge question answering (Zhang, 2020; Zhang and Yang, 2021b).

Subsequently, pre-trained models based on Transformer (Vaswani et al., 2017) like ELMo (Peters et al., 2018) or BERT (Devlin et al., 2018) are employed in a wide range of natural language processing tasks (Zhang and Yang, 2021a; Zhang et al., 2023). In question answering tasks, these pre-trained models are either combined with designed attention structures through embedding-style (Zhu et al., 2018; Zhang, 2019) or directly fine-tuned using concatenated historical dialogues.

However, regardless of the method used, the previous dialogue history needs to be encoded and interacted with in each turn of the conversation. It is not possible to save the previous dialogue states and directly use them.

## 3   Approaches

### 3.1   Task Formulation

In this section, we will illustrate our model from encoder to decoder. The task of the CQA can be formulated as follows. Suppose we are given a conversation, which contains a passage with $n$ tokens $P = \{w_t^P\}_{t=1}^n$ and multi-turn questions with $s$ question answering turns $Q = \{Q_r\}_{r=1}^s$. The $r$-th questions with $m$ tokens is $Q_r = \{w_{r,t}^Q\}_{t=1}^m$. And the model requires to give the corresponding answer $A_r$. A conversation in the dataset can be considered as a tuple $(P, Q, A)$.

### 3.2   Gated Recurrent Memory Reader

As shown in Figure 2, we use a generalized seq2seq framework to solve the task of CQA. As we know, the traditional seq2seq structure is used for many tasks, such as machine translation, semantic parsing, and so on. Given a sentence, the encoder will transform the input to the intermediate representation, which is used by the decoder to generate a new sentence word by word.

Similar to the seq2seq framework, our model also consists two modules: passage encoder and question decoder. the passage encoder module is designed to generate the hidden representation according to the passage. And question decoder module is for generating extractive answers turn by turn according to the result of the encoder.

There are also many differences between our model and seq2seq. First, the basic unit of our model is sentences rather than words. The intermediate representation (generated by the passage encoder) of our model contains the hidden states of all tokens in the passage. The input and the output of the decoder is also sentences. And the parallelism of our model is between conversations rather than sentences in a batch. Second, the output length of the question decoder depends on the turns of questions in a conversation. There is no start flag "$\langle GO \rangle$" or end flag "$\langle EOS \rangle$" in the decoder. Third, the output of the decoder, i.e., the answer, is not fed to the input of the next turn. Because the hidden states of the answer is in the memory of our model.

Our model can be formulated in Eq. 1. The decoder takes the result of the encoder $\bar{\mathbf{P}} = f_{enc}(\mathbf{P})$, and generates extractive answers according to questions.

$$
\begin{aligned}
P_\theta(\mathbf{A}|\mathbf{P},\mathbf{Q}) &= \prod_{j=1}^{s} P_{\theta_{model}}(\mathbf{A}_j|\mathbf{Q}_{\leq j},\mathbf{P}) \\
&= \prod_{j=1}^{s} P_{\theta_{decoder}}(\mathbf{A}_j|\mathbf{Q}_{\leq j},\bar{\mathbf{P}})
\end{aligned}
\quad (1)
$$

where $\mathbf{Q}_{\leq j} = \{\mathbf{Q}_1, \mathbf{Q}_2, \cdots \mathbf{Q}_j\}, j \in \{1, 2, \cdots s\}$. And $\mathbf{P}, \mathbf{Q}, \mathbf{A}$ is the embedding result of $P, Q, A$, respectively. For one conversation with $s$ question answering turns, $\mathbf{P} \in \mathbb{R}^{1 \times n \times h}$, $\mathbf{Q} \in \mathbb{R}^{s \times m \times h}$ and $\mathbf{A} \in \mathbb{R}^{s \times m \times h}$, where $h$ denotes the dimension of the embedding.

### 3.3 Passage Encoder Module

This module aims to encode the words of the passage into latent semantic representation, which will be used in the question decoder module. First, we obtain the embedding of the passage $e_t^{\text{GLV}}$ by the pre-trained Glove (Pennington et al., 2014). The part-of-speech (POS) and named entity recognition (NER) tags of each word are also transformed to the embedding vectors $e_t^{\text{POS}}$ and $e_t^{\text{NER}}$, respectively. They are learned during training. Then we concatenate them to a vector and feed it into the bi-directional recurrent neural network (RNN) to generate an intermediate representation of the passage $e_t^P = \text{BiRNN}(e_{t-1}^P, [e_t^{\text{GLV}}; e_t^{\text{POS}}; e_t^{\text{NER}}])$.

### 3.4 Question Decoder Module

This module is the core of our model. We take a turn of one conversation as an example to illustrate this module. And "turn" in our model is similar to "step" in RNN.

**Question Input Layer** Suppose the current question is the $r$-th question, we encode the embedding of the question $\{e_{r,i}^Q\}_{i=1}^m$ into $\{c_{r,i}^Q\}_{i=1}^m$ with a bi-directional RNN. We can obtain a vector of the question by weighted sum of tokens in Eq. 2. And $w_i$ represents different trainable weights in this section.

$$
c_r^{qsum} = \sum_{i=1}^{m} a_{r,i}^Q c_{r,i}^Q, \quad a_{r,i}^Q \propto exp(w_1^{\text{T}} c_{r,i}^Q) \quad (2)
$$

**Gated Memory Layer (Passage)** In this layer, we use a gated mechanism to leverage the information between the origin passage $e^P$ and the memory of the passage $c_{r-1}^P$, inspired by memory network(Kumar et al., 2016). The memory $c_{r-1}^P$ is obtained from the previous question turn. The $r$-th history-aware passage $c_r^P$ can be obtained as follows:

$$
\begin{aligned}
c_r^{sum} &= [c_{r-1}^{psum}; c_r^{qsum}] \\
g_r &= \sigma(w_2^{\text{T}} tanh(w_3^{\text{T}} c_r^{sum})) \\
c_r^P &= g_r c_{r-1}^P + (1 - g_r)e^P
\end{aligned}
\quad (3)
$$

where $c_{r-1}^{psum}$ is also obtained from the previous question turn. They will be interpreted in the next layer.

Specially, for the first question turn of the conversation, we directly use the intermediate representation, which is generated by the passage encoder, i.e., $c_r^P = e^P$.

**Interaction Layer** The passage and the $r$-th question will interact in this layer. First, we obtain the exact match feature $\hat{e}_{r,t}^{match}$, which indicate whether the token $w_t^P$ appears in the question $Q_r = \{w_{r,t}^Q\}_{t=1}^m$ in the form of prototype, lowercase and lemma. Then another match feature $\tilde{e}_{r,t}^{match}$ can be obtained by attention mechanism:

$$
\tilde{c}_{r,t}^{match} = \sum_{i=1}^{m} a_r^{i,t} e_{r,i}^Q, \quad a_r^{i,t} \propto exp(e_{r,i}^{Q\text{T}} e_t^P) \quad (4)
$$

We concatenate the information above for the passage and fed them to a bi-directional RNN to generate the question-aware passage representation $\bar{c}_{r,t}^P = \text{BiRNN}(\bar{c}_{r,t-1}^P, [c_{r,t}^P; \hat{e}_{r,t}^{match}; \tilde{e}_{r,t}^{match}])$. Then another attention is used to fuse the question and the passage as follows:

$$
h_{r,t}^{att} = \sum_{i=1}^{m} a_r^{i,t} c_{r,i}^Q, \quad a_r^{i,t} \propto exp(c_{r,i}^{Q\text{T}} \bar{c}_{r,t}^P) \quad (5)
$$

After that, we refine the representation of the passage by a bi-directional RNN, i.e., $\tilde{c}_{r,t}^P = \text{BiRNN}(\tilde{c}_{r,t-1}^P, [\bar{c}_{r,t}^P; h_{r,t}^{att}])$. Meanwhile, self-attention is also used to enhance the representation in Eq. 6. Then a bi-directional RNN integrates the representation above and generate the new $c_{r,t}^P$ in Eq. 7, which will be used by next turn in Eq. 3.

$$
h_{r,t}^{self} = \sum_{t=1}^{n} \tilde{a}_r^{i,t} \tilde{c}_{r,t}^P, \quad (6)
$$

$$
\tilde{a}_r^{i,t} \propto exp([h_{r,t}^{att}; c_{r,t}^P; \tilde{c}_{r,t}^P; \bar{c}_{r,t}^P]^{\text{T}}[h_{r,t}^{att}; c_{r,t}^P; \tilde{c}_{r,t}^P; \bar{c}_{r,t}^P])
$$

$$
c_{r,t}^P = BiRNN(c_{r,t-1}^P, [h_{r,t}^{self}; h_{r,t}^{att}; \tilde{c}_{r,t}^P]) \quad (7)
$$

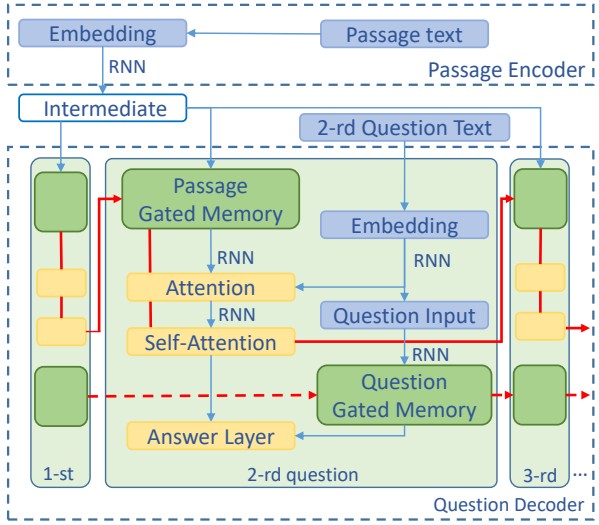

Figure 2: Overview of our model. (Best see in colors) (The red solid line refers to the memory flow of the passage. The red dashed line refers to the memory flow of the questions.)

| Model | F1 |
|---|---|
| Seq2seq | 27.5 |
| PGNet | 45.4 |
| DrQA | 54.7 |
| SDNet (w/o BERT lock weights) | 64.2 |
| DrQA+PGNet | 66.2 |
| **GRMR** | **66.3** |
| FlowQA (with ELMo) | 76.2 |
| SDNet (with BERT) | 77.7 |
| $MC^2$ (with BERT) | 81.3 |

Table 1: The performance on the CoQA dev set.

| Configuration | F1 | $\Delta$ F1 |
|---|---|---|
| GRMR | 66.3 | - |
| w/o gated memory (question) | 66.1 | -0.2 |
| w/o gated mechanism (passage) | 64.2 | -2.1 |
| w/o gated memory (passage) | 61.8 | -4.5 |
| w/o gated memory (all) | 58.8 | -7.5 |

Table 2: Ablation studies on the CoQA dev set.

Lastly, we can get the representation of the passage $c_r^{psum}$ by weighted sum of tokens like Eq. 2. It will be used both on gated memory layer in the next turn (in Eq. 3) and on gated memory layer for the current question (in Eq. 8). And the answer layer also uses the representation.

**Gated Memory Layer (Question)** As shown in Eq. 8, another gated memory is used to leverage the information of the current question $\hat{c}_r^{qsum}$ and the previous question memory $h_{r-1}^{qsum}$, where $\hat{c}_r^{qsum}$ is processed by a RNN cell, i.e., $\hat{c}_r^{qsum} = \text{RNNcell}(h_{r-1}^{qsum}, c_r^{qsum})$. Specialy, for the first question turn, we use the representation of the current question as the question memory, i.e., $h_r^{qsum} = c_r^{qsum}$. Then $h_r^{qsum}$ will be used in the answer layer and the next turn.

$$q_r^{sum} = [c_r^{psum}; c_r^{qsum}]$$
$$g_r = \sigma(w_4^T tanh(w_5^T q_r^{sum})) \quad (8)$$
$$h_r^{qsum} = g_r \hat{c}_r^{qsum} + (1 - g_r)h_{r-1}^{qsum}$$

**Answer Layer** This layer is the top layer of the question decoder module. Following pointer network (Vinyals et al., 2015) and DrQA (Chen et al., 2017), we use the bilinear function $f(x, y) = xWy$ to compute the probabilities of each token being start and end.

$$p_t^s \propto exp(f_s(c_{r,t}^P, c_r^{qsum})) \quad (9)$$

As shown in Eq. 9, we can obtain the start probability of each token $p_t^s$ in passage. And the end probability $p_t^e$ also can be obtained like this

by another bilinear function. We then concatenate the sentence-level representation of the passage and fed it to the linear classification $a_r = w_6^T[c_r^{psum}; h_r^{qsum}]$ for unanswerable questions.

## 4 Experiments

### 4.1 Dataset and Metric

We use the CoQA (Reddy et al., 2018) as our evaluation dataset. It is a large-scale conversational question answering dataset notated by people. It contains 127k questions with answers, obtained from 8k conversations about text passages from seven diverse domains. The text passages in the dataset are collected from seven diverse domains. And we use the F1 as the metric as the official evaluation.

### 4.2 Implementation Details

We use the Adamax (Kingma and Ba, 2014) as our optimizer. The initial learning rate is 0.004, and is halved after 10 epochs. To prevent overfitting, we set dropout to 0.4. The dimension of word embedding is 300, which is fixed during training. And the embedding dimension of POS and NER are set to 12 and 8, separately. we use LSTM as our recurrent neural network. The hidden size of LSTM is 128 throughout our model.

|         | 0-ctx | 1-ctx  | 2-ctx  | 3-ctx  |
|---------|-------|--------|--------|--------|
| train(KB) | 47854 | 183853 | 188960 | 193675 |
| multiple  | 1x    | 3.84x  | 3.95x  | 4.05x  |
| dev(KB)   | 8878  | 13008  | 13379  | 13723  |
| multiple  | 1x    | 1.47x  | 1.51x  | 1.55x  |

Table 3: Space occupancy on the CoQA.
(N-ctx refers to using previous N QA pairs)

### 4.3 Result

As shown in Table 1, we compare our method with other baseline models: PGNet(Seq2Seq with copy mechanism)(See et al., 2017), DrQA (Chen et al., 2017), DrQA+PGNet (Reddy et al., 2018), BiDAF++ (Yatskar, 2018), FlowQA (Huang et al., 2018), SDNet (Zhu et al., 2018) and MC$^2$ (Zhang, 2019). GRMR achieves comparable results to most models on the dev set, except FlowQA and SDNet. These two models all use the contextualized embedding, ELMo (Peters et al., 2018) or BERT (Devlin et al., 2018). However, our model still outperforms SDNet without BERT weights.

We also conduct ablation studies for our model in Table 2. We can find both the gated mechanism in memory and gated memory are crucial to our architecture. The score drops a lot without all gated memories. Different from the passage, the effect of the gated mechanism in question memory is not included, because our model can only use the current question without gate.

Lastly, we compare the storage space occupancy between our model and others on the CoQA dataset in Table 3. Our model takes up the least space with 0-ctx. Other models usually append 2 or 3 historical conversation turns to the current question. For the words in the training dataset, we can observe that the space they used is about four times larger than ours. And the difference will be larger when words are converted to vectors.

## 5 Conclusion

We propose a novel structure, GRMR, for conversational question answering. Traditional extractive MRC models are integrated into a generalized sequence-to-sequence framework. Gated mechanism and recurrent memory enable the model to consider the latent semantics of conversation history selectively and deeply with less space. The experiments show that this is a successful attempt to integrate extraction and generation in conversational question answering.

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
