# OpenReview forum: "Generating Extractive Answers: Gated Recurrent Memory Reader for Conversational Question Answering"
_EMNLP/2023/Conference — EMNLP 2023 Findings_

### Official Review · Reviewer_3NnP · 2023-08-04

**Soundness:** 2

**Excitement:**

2: Mediocre: This paper makes marginal contributions (vs non-contemporaneous work), so I would rather not see it in the conference.

**Paper Topic And Main Contributions:**

The paper prosed to incorporate Gated Recurrent Memory with traditional extractive MRC models to deal with conversational question answering task. Experiments on CoQA dataset proved the effectiveness and better space occupancy of proposed GRMR model, compared with traditional baselines such as DrQA, SDNet and PGNet.

**Questions For The Authors:**

Why not choose to use LLMs as base models?

**Reasons To Accept:**

A space-efficient gated memory mechanism to reduce space occupancy.

**Reasons To Reject:**

(1) The baselines involved in Table 1 are generally traditional non-LLM baselines before 2020, which are kind of out of date.
(2) Two LLM-equipped baselines in Table 1 are FlowQA (with ELMo) and SDNet (with BERT), which show quite better performance gaps beyond proposed GRMR model. What's more, the authors do not have a clear explanation on the improvement of space cost.

**Reproducibility:**

3: Could reproduce the results with some difficulty. The settings of parameters are underspecified or subjectively determined; the training/evaluation data are not widely available.

**Reviewer Confidence:**

4: Quite sure. I tried to check the important points carefully. It's unlikely, though conceivable, that I missed something that should affect my ratings.

---

> ### Author Rebuttal · Authors · 2023-08-29
>
> > (1) The baselines involved in Table 1 are generally traditional non-LLM baselines before 2020, which are kind of out of date. (2) Two LLM-equipped baselines in Table 1 are FlowQA (with ELMo) and SDNet (with BERT), which show quite better performance gaps beyond proposed GRMR model. What's more, the authors do not have a clear explanation on the improvement of space cost.
>
> > Why not choose to use LLMs as base models?
>
> Thank you for your question. The main focus of our paper was to propose a novel approach that integrates extractive MRC models with a gated recurrent memory mechanism for conversational question answering. Our goal was not to achieve state-of-the-art performance by leveraging existing LLMs, but rather to address the challenge of reducing space occupancy while maintaining competitive performance.
>
> Different from models based on the Transformer framework, including encoder (BERT), decoder (GPT), encoder-decoder (T5), our model adopts a structure similar to RNNs as the main framework. Training such a novel model architecture requires significant computational resources, extensive training data, and a substantial amount of time. For individual researchers, it is often not feasible to undertake such large-scale pretraining, which is necessary for LLMs. For example, a concurrent work, RWKV, requires extensive resources and retraining based on RNN-inspired architectures. Therefore, in Table 1, we included some results from LLM-based models to avoid missing important references. However, it is not a fair comparison as our model has significantly smaller scale (1/50~1/30000) and was not pretrained on massive corpora.
>
> In summary, the main contributions of our paper lie in the integration of extractive MRC models with a gated recurrent memory mechanism in the context of conversational question answering. We acknowledge that LLMs have become prominent in the field, and if we have the necessary resources in the future, we plan to explore their potential and conduct large-scale pretraining. We hope this explanation clarifies our approach and improves your evaluation of our paper.

---

### Official Review · Reviewer_wcpV · 2023-08-04

**Soundness:** 4

**Excitement:**

3: Ambivalent: It has merits (e.g., it reports state-of-the-art results, the idea is nice), but there are key weaknesses (e.g., it describes incremental work), and it can significantly benefit from another round of revision. However, I won't object to accepting it if my co-reviewers champion it.

**Missing References:**

The following reference/s also contain passage and answer with multi-turn questions.
* Raviteja Anantha, Svitlana Vakulenko, Zhucheng Tu, Shayne Longpre, Stephen Pulman, and Srinivas Chappidi. 2021. Open-domain question answering goes conversational via question rewriting. In Proceedings of the 2021 Conference of the North American Chapter of the Association for Computational Linguistics: Human Language Technologies, pages 520–534.

**Paper Topic And Main Contributions:**

Authors present Gated Recurrent Memory Reader (GRMR) which which integrates traditional extractive Machine Reading models into a generalized sequence-to-sequence framework. Authors show they achieve better space usage compared to several other models on CoQA dataset, but reports a wide gap with best performing models.

**Questions For The Authors:**

* Could you please update Table-3 to make it more granular, for example can you add separate rows for FlowQA and SDNet.
* A plot comparing trade-off in space vs F1 of GRMR vs best performing models would be very informative.

**Reasons To Accept:**

* Use of Gated Recurrent Memory Reader (GRMR) method for CQA seems to save space but w/ drop in F1 compared to best performing methods.

**Reasons To Reject:**

* The F1 Score obtained compared to other models, such as FlowQA and SDNet, on CoQA dataset has a wide gap.
* The improvement in space with 0-ctx doesn't seem proportionate to trade-off in F1.

**Reproducibility:**

2: Would be hard pressed to reproduce the results. The contribution depends on data that are simply not available outside the author's institution or consortium; not enough details are provided.

**Reviewer Confidence:**

4: Quite sure. I tried to check the important points carefully. It's unlikely, though conceivable, that I missed something that should affect my ratings.

---

> ### Author Rebuttal · Authors · 2023-08-29
>
> > The F1 Score obtained compared to other models, such as FlowQA and SDNet, on CoQA dataset has a wide gap.
>
> Thank you for your question. In Table 1, we included FlowQA and SDNet as references to avoid missing important comparisons. However, it is not a fair comparison as FlowQA and SDNet are both based on existing pretrained models. Our proposed GRMR model has a significantly smaller volume compared to these models and was not pretrained on massive corpora.
> The main contributions and innovations of our paper lie in integrating extractive MRC models with a gated recurrent memory mechanism and achieving better space occupancy in the context of conversational question answering. Our goal was not to achieve state-of-the-art performance by leveraging existing LLMs and surpassing other models.

---

### Official Review · Reviewer_G5eK · 2023-08-05

**Soundness:** 4

**Excitement:**

4: Strong: This paper deepens the understanding of some phenomenon or lowers the barriers to an existing research direction.

**Paper Topic And Main Contributions:**

The authors propose a new RNN architecture for Conversational Question Answering tasks. Their model comprises 2 modules, 1) Passage encoder, 2) Question decoder. The question decoder contains a question-passage attention, self-attention, and Gated Memory Layer to encoder sequences of conversational questions in a memory-efficient way. Experiments on CoQA dataset show that their model beats most of the baselines except for models using contextualized embeddings such as Elmo and BERT.

**Questions For The Authors:**

A - In line 115, the authors mentioned "the output of the decoder, i.e.,the answer, is not fed to the input of the next turn". Won't this lead to problems if the subsequent question is referring to the preceding answer string?

**Reasons To Accept:**

- new memory efficient architecture for Conversational QA task

**Reasons To Reject:**

- Achieves 10 points less F1 than the standard BERT model architecture. I am assuming that the performance gap will be even higher when using more recent and better-pretrained models such as RoBERTa, GPTs, LLAMAs etc.
- Lacking memory overhead comparisons between baselines (there should be a table in the appendix at least)

**Reproducibility:**

5: Could easily reproduce the results.

**Reviewer Confidence:**

5: Positive that my evaluation is correct. I read the paper very carefully and I am very familiar with related work.

---

> ### Author Rebuttal · Authors · 2023-08-29
>
> > Achieves 10 points less F1 than the standard BERT model architecture. I am assuming that the performance gap will be even higher when using more recent and better-pretrained models such as RoBERTa, GPTs, LLAMAs etc. Lacking memory overhead comparisons between baselines.
>
> Thank you for your comments. Incorporating large-scale pretrained models can indeed improve model performance. However, the main contributions and innovations of our paper revolve around integrating extractive models into generative models, achieving reduced space occupancy in multi-turn conversations. Our goal was not to simply integrate and rely on existing LLMs for achieving state-of-the-art performance.
>
> Moreover, due to structural differences, many LLMs cannot be directly integrated and would require retraining with massive amounts of data. Our proposed approach offers a powerful and influential idea that can be extended to other models and tasks.
> Additionally, it's worth noting that the memory usage of integrating pretrained language models is significantly higher compared to our model (approximately 50x-30000x higher).
>
> > In line 115, the authors mentioned "the output of the decoder, i.e.,the answer, is not fed to the input of the next turn". Won't this lead to problems if the subsequent question is referring to the preceding answer string?
>
> This is a great question. We have actually conducted experiments on whether to include the output of each round as input for the next round, and overall, the results show that not including the output yields better performance. Intuitively, this can be seen as a question of explicit versus implicit information transfer. Dense intermediate vectors can effectively carry forward the implicit states from previous rounds.

---

### Meta-Review · Area_Chair_vASm · 2023-09-26

**Recommendation:** 3

**Metareview:**

This paper presents a new architecture that provides direct access to passage rep and question history for conversational QA, through a gated memory layer. This provides inference-time space savings that become increasingly significant as the conversation grows.

The new approach achieves significantly worse performance than recent architectures that use contextualized embeddings. However, those also require much more memory to run. Compared to similar base architectures, the new approach performs well although it should be noted that the closest comparison from Reddy et.al 2018 achieves essentially the same performance, in terms of accuracy.

Multiple reviewers raised questions about the significant gap in performance to more recent, Transformer-based architectures and whether the accuracy / space trade-off is reasonable, especially for the 0-ctx case. It is clear that the current focus of the paper is not just increasing the accuracy metrics but the work would benefit from a deeper discussion of these tradeoffs and / or discussion of how they could be overcome.

---

### Decision · Program_Chairs · 2023-10-07

**Decision:**

Accept-Findings

**Comment:**

This paper presents a new architecture that provides direct access to passage rep and question history for conversational QA, through a gated memory layer. This provides inference-time space savings that become increasingly significant as the conversation grows.

The new approach achieves significantly worse performance than recent architectures that use contextualized embeddings. However, those also require much more memory to run. Compared to similar base architectures, the new approach performs well although it should be noted that the closest comparison from Reddy et.al 2018 achieves essentially the same performance, in terms of accuracy.

Multiple reviewers raised questions about the significant gap in performance to more recent, Transformer-based architectures and whether the accuracy / space trade-off is reasonable, especially for the 0-ctx case. It is clear that the current focus of the paper is not just increasing the accuracy metrics but the work would benefit from a deeper discussion of these tradeoffs and / or discussion of how they could be overcome.